# Efficiency of Different Supplements in Alleviating Symptoms of ADHD with or Without the Use of Stimulants: A Systematic Review

**DOI:** 10.3390/nu17091482

**Published:** 2025-04-28

**Authors:** Shatha Al Shahab, Rawan Al Balushi, Amna Qambar, Ruqayah Abdulla, Maryam Qader, Shooq Abdulla, Haitham Jahrami

**Affiliations:** 1College of Medicine and Medical Sciences, Arabian Gulf University, Manama P.O. Box 26671, Bahrain; rawanrhb@agu.edu.bh (R.A.B.); amnawsq@hotmail.com (A.Q.); ruqayahmma@agu.edu.bh (R.A.); maryammaq@agu.edu.bh (M.Q.); shooqfaa@agu.edu.bh (S.A.); 2Governmental Hospitals, Manama P.O. Box 12, Bahrain

**Keywords:** attention deficit hyperactivity disorder, ADHD, inattention, hyperactivity, impulsivity, L-theanine, caffeine, *Ginkgo biloba* L., *Bacopa monnieri*, systematic review

## Abstract

Objective: This review aims to assess the efficacy and safety of different supplements, such as L-theanine, caffeine, *Ginkgo biloba* L., and *Bacopa monnieri* for improving ADHD symptoms, to determine the most effective supplement and provide insight for medical practice. Methods: International databases (PubMed/MEDLINE and Scopus) were searched for English-language RCTs, open-label studies, and cross-sectional studies. Results: Studies on L-theanine, caffeine, *Ginkgo biloba* L., and *Bacopa monnieri* have shown various effects on ADHD symptoms. L-theanine improved sleep efficiency but not other sleep parameters. Caffeine showed no significant benefits, although its combination with L-theanine may enhance attention. *Bacopa monnieri* consistently improved inattention, hyperactivity, and memory, whereas *Ginkgo biloba* L. reduced ADHD symptoms, particularly inattention, but was less effective than methylphenidate. Conclusions: The evidence revealed the potential use of supplements as complementary ADHD treatments under clinical guidance. The limited effect of supplements cannot replace the well-documented efficacy of stimulants for ADHD treatment. Additional studies are needed to ascertain the most effective dosages and the safety of these supplements as adjunctive treatments for ADHD.

## 1. Introduction

Attention deficit hyperactivity disorder (ADHD) is considered one of the most common neurodevelopmental disorders that present during childhood, with onset occurring before 7 years of age. Children with ADHD suffer from inattentiveness, hyperactivity, and impulsivity. All of these factors can result in functional impairment in terms of school performance, such as the inability to finish given tasks or follow instructions [1]. In addition, both social relationships and family functioning can be negatively affected by behaviors such as fidgeting, excessive talking, and inability to sit through a whole meal [2]. Although the causative agents are unknown, possible risk factors have been identified. These factors include genetics, exposure to alcohol and tobacco during fetal development, and the familial environment.

The treatment of ADHD depends on the affected child’s age, as it is a combination of lifestyle modifications, behavioral therapy, and pharmacotherapy [3]. Dietary changes and interventions, getting the recommended amount of sleep, and engaging in physical activities are all considered part of the modifications. Behavioral therapy is performed for children, as well as behavioral parent training, in which building structure, maintaining discipline, and using positive reinforcement help parents effectively manage their child’s negatively presumed behavior. Finally, pharmacotherapy using stimulants such as methylphenidate and amphetamines is more effective when combined with behavioral therapy. Nonstimulants (atomoxetine, guanfacine, and clonidine) are considered secondary treatment options. Several studies have investigated the effects of supplements, such as L-theanine, caffeine, *Ginkgo biloba* L., and *Bacopa monnieri* on alleviating symptoms of ADHD, all of which have shown varied results [4].

L-theanine is an amino acid primarily found in green and black tea and certain types of mushrooms. It is acknowledged for its ability to improve cognitive function, predominantly attention [5]. Caffeine is a naturally derived compound that has stimulant properties. It is predominantly found in plant sources such as coffee beans and tea leaves, as well as artificially added to soft drinks such as cola. Caffeine has been widely utilized for decades to increase sustained attention, benefiting both healthy adults and individuals with ADHD [6]. Furthermore, *Ginkgo biloba* L. is widely and commonly found in China. It is known for its antioxidant properties, neurotransmitter/receptor modulatory functions, and ability to inhibit platelet activation. All of these factors have the potential to improve memory, cognition, stroke, Alzheimer’s disease, and aging [4]. Finally, *Bacopa monnieri* is an herbal Ayurvedic medicine that has been used in the past 3000 years for its ability to improve cognition and memory and promote longevity [7].

The incorporation of natural supplements in the management of ADHD, often referred to as “nutraceuticals”, involves three primary mechanisms: addressing deficiencies (e.g., replenishing essential nutrients critical for neurotransmitter function), supplementing physiological processes (e.g., enhancing attention through the stimulant properties of caffeine), and administering supraphysiological doses to achieve therapeutic effects (e.g., high-dose *Bacopa monnieri* extracts to enhance cognitive function). The supplements reviewed—L-theanine, caffeine, *Bacopa monnieri*, and *Ginkgo biloba* L.—primarily fall into the latter two categories, with the aim of augmenting existing pathways or modulating symptoms through concentrated bioactive compounds.

The aim of this systematic review is to employ relevant available literature to assess the effectiveness and safety of the commonly used supplements L-theanine, Caffeine, *Ginkgo biloba* L., and *Bacopa monnieri* in improving ADHD symptoms and to gain a comprehensive understanding of their use in medical practice. In addition, we aim to determine which supplement has greater efficacy and a superior profile in alleviating symptoms in ADHD patients. Therefore, this study provides valuable insights for current medical practice and contributes to future medical research.

## 2. Materials and Methods

### 2.1. Study Design

This review was conducted and reported in accordance with the Synthesis without Meta-analysis (SWiM) guidelines to ensure methodological rigor and transparency [8]. The SWiM guidelines complement and extend the PRISMA 2020 guidelines, providing additional structure for synthesizing evidence without meta-analysis. Both checklists are available in the Appendix A. Four of the team members independently conducted searches across two online databases from their time of publication until 10 February 2025. The purpose of this study was to assess the efficacy of various supplements (L-theanine, Caffeine, *Ginkgo biloba* L., and *Bacopa monnieri*) in alleviating ADHD symptoms, both with and without the use of stimulants. Two electronic databases, PubMed/MEDLINE and Scopus, were searched using relevant keywords, including “L-theanine”, “Caffeine”, “*Ginkgo biloba*”, “Bacopa”, and “ADHD”.

### 2.2. Search Strategy and Selection Criteria

The selected studies were required to fulfill the following predefined criteria: (i) study design: studies that provided relevant data on the use of amino acids and/or herbal supplements in ADHD patients; (ii) participants: humans of all ages and genders; (iii) main intervention: studies investigating the use of specific supplements, including L-theanine, caffeine, *Ginkgo biloba* L., and *Bacopa monnieri*; and (iv) language: studies published in English. Studies meeting any of the following conditions were excluded: (i) animal studies; (ii) studies focusing on military personnel or athletes; (iii) studies examining different supplement combinations beyond those specified in the inclusion criteria; (iv) studies involving patients diagnosed with other psychological disorders; (v) studies for which full texts were not available; and (vi) systematic reviews and meta-analyses. After removing duplicate records from the selected online databases, a primary screening of article titles and abstracts was performed. A review of the full texts of the eligible articles was then conducted by S.A.S., R.A.B., A.Q., and R.A. individually. Only randomized, double-blind, placebo-controlled, cross-sectional, and open-label clinical trials that reported the use of amino acid and herbal supplements in ADHD patients were selected for final inclusion in the review. The study screening and selection process is illustrated via a PRISMA flow diagram (Figure 1).

### 2.3. Data Extraction

Data from the selected studies were extracted using a predefined table that included the following: authors, study design, country, duration, year of publication, participant characteristics (age, sex, sample size), intervention (supplement, dose, with or without stimulant), scales, and findings (whether the supplement improved the symptoms of ADHD or not) (Table 1).

### 2.4. Quality Assessment

The “Cochrane Collaboration’s Tool for Assessing Risk of Bias” [9] was used for the bias assessment of controlled trials among the selected studies. It includes five domains: the randomization process, deviations from the intended interventions, missing outcome data, measurement of the outcomes, and selection of reported results. Each study was assessed for overall quality and categorized as having a low risk of bias, some concerns regarding bias, or a high risk of bias, as presented in Figure 2.

**Table 1 nutrients-17-01482-t001:** Characteristics of the included studies.

Author/Year/ Country	Study Design/ Duration	⁠Characteristics (Sample Size, Age, Gender)	Supplement/Dose	Scales	Results
Kahathuduwa et al., 2020—USA [10]	Randomized placebo-controlled four-way repeated measures crossover trial—14 May 2018–31 August 2018	5 Male gender 8–15 years	Caffeine (2.0 mg/kg)/L-theanine (2.5 mg/kg). Four participants were not on stimulant medications. One participant used a dose of methylphenidate hydrochloride as needed.	1 h post-dose: Go/NoGo task and a Stop-signal task 2 h post-dose: NIH Cognition Toolbox	Evident sustained attention, inhibitory control, and overall cognitive performance.
Lyon et al., 2011—Canada [11]	Randomized, double-blind, placebo-controlled trial—10 weeks	98 Male gender 8–12 years	L-theanine (100 mg): two chewable tablets twice daily, morning and afternoon (total 400 mg daily). Equal distribution of stimulant- and non-stimulant-treated participants.	Objective (actigraphy) and subjective (PSQI) measures	Increased sleep percentage and sleep efficiency scores, insignificant trend for less activity during sleep. No difference in sleep latency and other sleep parameters.
Ágoston et al., 2022 [12]	Cross-sectional study—NA	2259 70% males, 30% females > 18 years	Caffiene: daily average for males: 255.40 mg; daily average for females: 223.35 mg	Caffeine Use Disorder Questionnaire (CUDQ) Adult ADHD Self-report Scale (ASRS) WHO-5 Well-Being Index (WHO-5)	No relation between caffeine consumption and ADHD symptom severity
Arnold, 1978 [13]	Randomized placebo-controlled crossover trial—21 days	29 M = 22, F = 7 5–12 years	Caffiene (160–300 mg)	Conners’ Teachers Rating Scale	Not significantly better than placebo
Firestone, 1978 [14]	Randomized placebo-controlled crossover trial—21 days	21 Gender distribution NA 6–12 years	Caffiene (300–500 mg)	Matching Familiar Figures Test (MFF) The Maze Test Conners’ Parent Rating Scale Conners’ Teacher Rating Scale Conners’ Short Form Rating Scale	No difference between placebo and caffeine.
Garfinkel, 1975 [15]	Randomized placebo-controlled crossover trial—10 days	8 Gender distribution NA 6–10 years	Caffiene (150 mg)	Conners’ Teacher Rating Scale Bender Visual Motor Gestalt Test Frostig Developmental Test of Visual Perception (part II and part IV) Reitan Neuropsychological Battery Test for Motor Coordination and Steadiness Kagan Matching Familiar Figures Test	Caffeine did not significantly improve scores on any of the scales
Huestis, 1975 [16]	Randomized placebo-controlled crossover trial—21 days	18 M = 12, F = 6 5–12 years	Caffiene (80–300 mg)	Davids Hyperkinetic Rating Scale Conner’s Teachers Rating Scale	Not significantly better than placebo
Kean et al., 2022—Australia [17]	Randomized, double-blind, placebo-controlled clinical trial—14 weeks	112 Male gender 6–14 years	*Bacopa monnieri* extract (CDRI 08^®^)	NA	Significant improvements were observed in inattention, hyperactivity, and cognitive function in the treatment group compared to the placebo group.
Kean et al., 2015—Australia [18]	Randomized, placebo-controlled, double-blind, parallel-group trial—16 weeks	100 Male gender 6–14 years	*Bacopa monnieri* extract (CDRI 08) (160 mg capsule)	Conners’ Parent Rating Scale (CPRS)	Improvement in symptoms of inattention, hyperactivity, and impulsivity
Dave et al., 2014—India [19]	Open-label clinical trial—6 months	27 M = 24, F = 3 6–12 years	*Bacopa monnieri* extract (225 mg/day)	NA ADHD symptoms subtest scores	Improvements in restlessness in 93% of participants, impulsivity in 67%, inattention in 85%, self-control in 89%, psychiatric problems in 52%, and learning problems in 78%
Dave et al., 2008—India [20]	Open-label clinical trial—4 months	28 M = 13, F = 15 4–18 years	*Bacopa monnieri* extract (225 mg/day)	Memory Scale Test	Significant improvement in working memory, logical memory, personal life memory, short-term verbal memory, and visual and auditory memory.
Shakibaei et al., 2015—Iran [21]	Randomized, placebo-controlled, double-blind clinical trial—6 weeks	60 Gender distribution NA 6–12 years	*Ginkgo biloba* L. (80–120 mg/day)	ADHD Rating Scale-IV (ADHD-RS-IV)	The treatment group showed a greater reduction in inattention, compared to the placebo group. The treatment group demonstrated a higher response rate based on parent ratings, compared to the placebo group (93.5% vs. 58.6%).
Sandersleben, 2014—Germany [22]	Open clinical pilot study—3–5 weeks	20 M = 15, F = 5 6–13 years	*Ginkgo biloba* L.: initial dose increased gradually up to a maximum of 240 mg	Conner’s Continous Performance Test Contingent Negative Variation Test	Improvement in ADHD core symptoms, improvement in quality of life and overall performance
Salehi et al., 2010—Iran [23]	Double-blind, randomized clinical trial—6 weeks	50 M = 39, F = 11 6–17 years	*Ginkgo biloba* L. (80–120 mg/day), or methylphenidate	NA	Methylphenidate was significantly more effective than *Ginkgo biloba* L. in reducing ADHD symptoms. Less side effects, such as decreased appetite, headache, and insomnia, in the *Ginkgo biloba* L. group compared to the methylphenidate group.

Observational and open-label studies were evaluated using the Risk of Bias in Nonrandomized Studies of Interventions (ROBINS-I) tool. Each study was assessed across seven domains: bias from confounding, selection of participants, classification of interventions, deviations from intended interventions, bias from missing data, measurement of outcomes, and selection of reported results. The results are classified as having a low, moderate, or serious risk of bias, as detailed in Figure 3.

**Figure 2 nutrients-17-01482-f002:**
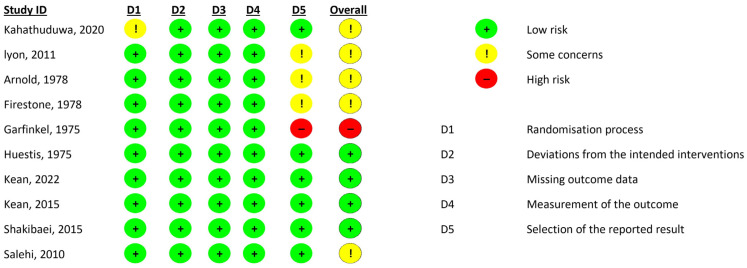
Risk of bias assessment of controlled studies [10,11,13,14,15,16,17,18,21,23] using Cochrane Collaboration’s tool.

### 2.5. Ethics

As this systematic review analyzed publicly available data, ethical approval was not required.

## 3. Results

A total of 372 records were obtained from the selected databases, with 178 from PubMed/MEDLINE and 194 from Scopus. Fifty-one duplicate records were eliminated, and 321 titles and abstracts were screened, leading to 46 studies eligible for full-text review. Google Scholar was used to cross-search for full texts. However, 12 of these studies were inaccessible. Consequently, 34 studies were assessed for eligibility, with 20 being excluded primarily because they did not meet the specified study design criteria, as shown in Figure 1. Ultimately, 14 studies were included in the research, as shown in Table 1.

### 3.1. L-Theanine

Our research investigated the effects of L-theanine on individuals diagnosed with ADHD, identifying two relevant studies. The first study, conducted by Kahathuduwa [10], investigated the effects of L-theanine and caffeine together on cognitive performance, attention management, and impulse control in five ADHD-diagnosed boys aged 8–15 years. These findings suggest that this combination could serve as a therapeutic intervention for ADHD-related cognitive challenges. The study revealed a statistically significant improvement in total cognition composite scores within the NIH Cognition Toolbox (*p* = 0.040) when L-theanine was taken versus a placebo. However, inhibitory control worsened when L-theanine and caffeine were administered separately, as indicated by increased stop signal reaction times (*p* = 0.031 and *p* = 0.053, respectively). The results indicated that this combination enhanced total cognition composite scores (*p* = 0.041) and d-prime scores in the Go/No-Go task (*p* = 0.033) while also resulting in a trend toward better inhibitory control (*p* = 0.080). Additionally, it was linked to decreased task-related reactivity in the default mode network, which is associated with mind wandering.

The second study, conducted by Lyon [11], assessed the safety and efficacy of L-theanine in improving the quality of sleep in 98 boys aged 8–12 years who were diagnosed with ADHD. The participants were divided into two groups: 46 received 400 mg of chewable L-theanine tablets daily, consisting of two 100 mg tablets in the morning and two 100 mg tablets in the late afternoon, whereas 47 were given a placebo.

Sleep quality was assessed via both objective (actigraphy) and subjective (PSQI) measures. These findings indicate that L-theanine enhances sleep quality in children with ADHD, as evidenced by increased sleep percentage and efficiency, together with a tendency toward reduced sleep activity. However, it did not significantly affect the onset or duration of sleep. There was a slight trend indicating a shorter wake time after sleep onset in the L-theanine group. The intervention demonstrated excellent tolerability, with no serious adverse events reported. A single mild adverse event (transient facial tic) occurred in a participant with a pre-existing tic disorder, which the investigators concluded was unrelated to treatment. Overall, L-theanine seems to be a safe and effective option for improving sleep quality in male children with ADHD and has the potential as an additional treatment.

### 3.2. Caffeine

Caffeine was one of the supplements used in this study. Five studies met the systematic review criteria for inclusion. Agoston et al. [12] tested daily caffeine consumption and its effect on ADHD symptoms in adults. This was a cross-sectional study including 2259 participants, with an average daily caffeine consumption of 255.4 mg for males and 223.35 mg for females. The 10-item Caffeine Use Disorder Questionnaire (CUDQ), the WHO-5 Well-Being Index (WHO-5), and the Adult ADHD Self-report Scale (ASRS) were used to evaluate ADHD symptoms; as a result, the intensity of ADHD symptoms was not found to be correlated with caffeine use. Additionally, Arnold [13] conducted a randomized placebo-controlled crossover trial that concluded after 21 days and tested caffeine doses ranging from 160 to 300 mg using the Conners’ Teacher Rating Scale as an assessment tool to determine the effectiveness of caffeine on ADHD symptoms; the results revealed that caffeine did not improve ADHD symptoms.

Another randomized placebo-controlled crossover trial performed by Firestone [14] over 21 days involved 21 children aged 6 to 12 years. The children received caffeine doses varying from 300 to 500 mg per day, and outcomes were measured via the Matching Familiar Figures Test (MFF) and the Conners Rating Scale. Consequently, the results revealed no improvement in the cognitive or behavioral symptoms of ADHD in the caffeine group compared with the placebo group.

An earlier study in 1975, investigated the effect of a low dose of caffeine (150 mg daily) on 8 children aged 6 to 10 years over 10 days [15]. A wide range of scales, including the Conners’ Teacher Rating Scale, the Bender Visual Motor Gestalt Test, the Frostig Developmental Test of Visual Perception (parts II and IV), the Reitan Neuropsychological Battery Test for Motor Coordination and Steadiness, and the Kagan Matching Familiar and Figures Test, were used. The findings revealed that low doses of caffeine had no significant effect on ADHD symptoms and that there was no improvement in any of the tested factors.

In 1975, one of the earliest studies of the effect of caffeine on children with ADHD investigated how caffeine affects symptoms of ADHD [16]. Eighteen children between the ages of five and twelve, including 12 males and 6 females, were included in a randomized, placebo-controlled crossover study over 21 days. Numerous rating measures were used to assess the children’s responses to daily caffeine dosages ranging from 80 to 300 mg. These measurements included the Davids Hyperkinetic Rating Scale, the parent symptom checklist, the Conners’ Teacher Rating Scale, and the Behavior Checklist—Target Symptom Assessment by parents with psychiatrist assistance. The findings demonstrated that caffeine did not significantly improve ADHD symptoms compared with a placebo.

### 3.3. Bacopa monnieri

For the supplement *Bacopa monnieri*, four studies investigating the influence of *Bacopa monnieri* on ADHD symptoms met the inclusion criteria for inclusion in the systematic review [17,18,19,20]. A randomized, double-blind, placebo-controlled clinical trial was conducted by Kean et al. [17] in Australia, involving a population of 112 males displaying symptoms of inattention and hyperactivity aged 6–14 years. The results revealed significant improvements in inattention, hyperactivity, and cognitive function in the group receiving compared with the placebo group, the *Bacopa monnieri* group performed better on tasks related to attention and hyperactivity.

In an earlier randomized, placebo-controlled, double-blind, parallel-group trial, Kean et al. [18] investigated the impact of *Bacopa monnieri* (CDRI 08)/160 mg capsules on hyperactivity and inattention symptoms in 100 male children and adolescents ranging from 6 to 14 years over a span of 16 weeks. Using the Conners’ Parent Rating Scale (CPRS), EEG, cognition, mood, and sleep assessments, *Bacopa monnieri* improved hyperactivity, impulsiveness, and distractibility symptoms in children.

An open-label clinical trial lasting 6 months that included 3 males and 24 females ranging from 6 to 12 years was carried out by Dave et al. in India [19]. The aim of the trial was to use ADHD symptom subtest scores to measure the outcomes of *Bacopa monnieri* (225 mg/d) in managing symptoms of ADHD in children. The results revealed improvements in restlessness (93%), impulsivity (67%), inattention (85%), self-control (89%), psychiatric problems (52%), and learning problems (78%).

The fourth study included an open-label clinical trial that evaluated the effects of *Bacopa monnieri* (225 mg/d) on intellectual and memory functions in children aged 4 to 18 years, including 13 males and 15 females, via memory scale tests [20]. The trial lasted for 4 months and revealed significant improvements in working memory, immediate verbal memory, logical recall, autobiographical memory, and visual and auditory recall.

### 3.4. Ginkgo biloba L.

Three studies supplemented with *Ginkgo biloba* L. met the inclusion criteria for this systematic review [21,22,23]. Shakibaei et al. [21] carried out a randomized, placebo-controlled trial in Iran that lasted 6 weeks. They included children aged 6 to 12 years to assess the efficacy of *Ginkgo biloba* L. (80–120 mg/day) as an add-on therapy for attention-deficit/hyperactivity disorder via the ADHD Rating Scale-IV (ADHD-RS-IV). The results revealed that the group receiving *Ginkgo biloba* L. presented a greater reduction in ADHD-RS-IV parent-rated inattention scores (−7.74 ± 1.94) than did the placebo group (−5.34 ± 1.85), with a highly significant difference (*p* < 0.001). In addition, the *Ginkgo biloba* L. group presented a higher response rate on the basis of parent ratings (93.5% vs. 58.6%, *p* = 0.002).

Another open clinical pilot study, carried out by Sandersleben [22] in Germany, investigated the relationship of *Gingko biloba* (EGb 761^®^) with the brain’s electrical activity and its efficacy in children diagnosed with ADHD combined with the DSM-IV. The study included 20 children, 75% of whom were boys aged 6–13 years. In this study, if the patient’s symptoms continued despite the use of the supplement *Ginkgo biloba* L., the dosage of the supplement would be increased by only 240 mg daily. Efficacy was evaluated by assessing quality of life (QoL), and performance and preparatory brain-electrical activity were measured during a continuous performance test (Cue-CNV in the CPT). The results revealed improvements in ADHD core symptoms, quality of life, and performance.

The last study measured the effectiveness of *Ginkgo biloba* L. by carrying out a double-blind, randomized, parallel-group comparison of *Gingko biloba* (80–120 mg/day) and methylphenidate [23]. The study included 50 children (11 girls and 39 boys) aged 6 to 17 years. The results of this study revealed that methylphenidate was significantly more effective than *Ginkgo biloba* L. in reducing ADHD symptoms. However, the *Ginkgo biloba* L. group reported fewer adverse symptoms, such as reduced appetite, insomnia, and headaches, than did the methylphenidate group.

## 4. Discussion

This systematic review aimed to assess the efficacy of various herbal and amino acid supplements in improving symptoms of ADHD. More specifically, we focused on the effects of L-theanine, caffeine, *Bacopa monnieri*, and *Ginkgo biloba* L. The studies included in this review highlight both beneficial outcomes and those with mixed or vague results. This, in turn, offers insight into the potential role of these supplements in ADHD management.

L-theanine has beneficial effects on sleep, with a study by Lyon et al. [11] showing that L-theanine supplementation resulted in increased sleep rates and improved sleep quality scores. On the other hand, the same results did not indicate any significant improvement in sleep latency or other parameters. These findings suggest that while L-theanine may have beneficial effects on sleep quality, its impact on ADHD-related symptoms may be limited to certain aspects, particularly sleep, rather than core symptoms such as inattention or hyperactivity.

Similarly, caffeine’s effects have also been tested in several studies but have shown inconsistent results. Kahathudwa et al. [10] reported that caffeine, when combined with L-theanine, improved sustained attention, response inhibition, and cognitive function. However, caffeine-only supplementation studies, such as those by Arnold [13], Firestone [14], Garfinkel [15], and Huestis [16] reported no significant effects when caffeine was compared to a placebo in relation to ADHD symptoms. These mixed and inconsistent findings suggest that while caffeine may have some cognitive benefits, its efficacy as a treatment for ADHD remains uncertain, and it may not significantly improve core symptoms when used alone.

In contrast, *Bacopa monnieri* appeared to be the supplement with the most consistent effects. Several studies, including those by Kean et al. [17,18] and Dave et al. [19,20], reported significant improvements in ADHD symptoms, specifically inattention, hyperactivity, and impulsivity. *Bacopa monnieri* was particularly effective in enhancing cognitive function and attention, with Dave et al. [19] reporting an impressive 85% improvement in inattention. These findings align with previous research and further support *Bacopa monnieri*’s potential in improving both cognitive performance and symptom control in individuals with ADHD, highlighting its promising role in ADHD management.

The results for *Ginkgo biloba* L. were inconclusive. Shakibaei et al. [21] reported a significant reduction in ADHD-RS-IV parent-rated inattention scores in the *Ginkgo biloba* L. group compared with the placebo group, which suggests a potential improvement in inattention. Additionally, Sandersleben [22] demonstrated improvements in different aspects, including ADHD core symptoms, quality of life, and continuous performance testing. However, a study by Salehi et al. [23] revealed that while *Ginkgo biloba* L. was less effective than methylphenidate in reducing ADHD symptoms, it resulted in fewer side effects, including decreased appetite, headache, and insomnia. This finding may suggest that *Ginkgo biloba* L. could be a suitable alternative for individuals who experience significant side effects from traditional ADHD medications.

### Strengths and Limitations

A notable strength of this review is its ability to provide a comprehensive overview of the available evidence on the efficacy of herbal and amino acid supplements for managing and improving ADHD symptoms through a systematic review approach.

While the reviewed supplements—L-theanine, caffeine, *Bacopa monnieri*, and *Ginkgo biloba* L.—demonstrate varying degrees of efficacy in alleviating ADHD symptoms, their regulatory status as non-pharmaceutical products must be critically considered. Globally, these supplements undergo less stringent scrutiny than conventional medications, raising concerns about standardized dosing, quality control, and long-term safety. The variability in supplement quality, composition, and bioavailability poses significant challenges for interpreting the efficacy and generalizability of findings in studies involving herbal and amino acid interventions for ADHD. While L-theanine and caffeine are chemically simpler than whole plant extracts, their formulations remain inconsistent due to regulatory gaps. Manufacturers may standardize these compounds to ensure stability and bioavailability, but products labeled as “L-theanine” or “caffeine” can vary in purity, dissolution rates, or additive ingredients, potentially influencing outcomes. For instance, the modest benefits of L-theanine on sleep efficiency observed in [11] and the synergistic effects of caffeine-L-theanine combinations in [10] may not be replicable across all formulations, particularly if manufacturing practices differ. This underscores the need for rigorous quality control standards, even for ostensibly “simple” supplements.

A critical limitation of many trials in this review is the lack of transparency regarding batch consistency. Unless a study explicitly uses a single batch of a standardized extract (as in [17]), assumptions about consistent dosing across participants and over time remain speculative. This variability complicates comparisons between studies and undermines the reliability of meta-analyses or clinical recommendations. For instance, the open-label *Bacopa monnieri* trials by Dave et al. [19,20] reported improvements in ADHD symptoms and memory, but without detailed documentation of extract standardization or batch testing, these results may not extend to other formulations. Similarly, the *Ginkgo biloba* L. trials reviewed in [21,23] used extracts with unspecified phytochemical profiles, raising questions about reproducibility.

The challenges are further amplified for whole plant extracts such as bacopa monnieri and *Ginkgo biloba* L., which contain hundreds of bioactive compounds whose concentrations depend on environmental, agricultural, and extraction variables. For example, *Bacopa monnieri* studies like those by Kean et al. [17,18] and Dave et al. [19] utilized standardized extracts (e.g., CDRI 08^®^) to partially mitigate variability by controlling key active constituents like bacosides. However, even these efforts cannot account for the full complexity of plant-derived formulations. Extraction methods—such as solvent type, temperature, and pH—can alter the chemical profile of the final product, as seen in *Ginkgo biloba* L. studies where differences in terpene lactones or flavonoid content might influence efficacy [21,23]. The WHO’s emphasis on documenting medicinal plants within their native “terroir” highlights the importance of contextual factors (e.g., cultivar, climate, harvest timing) that are rarely replicated in commercial production.

These limitations necessitate cautious interpretation of the evidence. While *Bacopa monnieri* and *Ginkgo biloba* L. show promise as adjunctive therapies, their efficacy and safety profiles cannot be generalized to all products labeled as such. Future research must prioritize the standardization of extracts, transparency in sourcing and manufacturing practices, and stricter regulatory oversight to ensure consistency and safety. Clinicians should exercise vigilance in recommending these interventions, advising patients to prioritize third-party-verified products and use supplements only under medical supervision. The findings herein reinforce the need for large-scale, methodologically rigorous trials to address these complexities and establish reliable, evidence-based guidelines for ADHD management.

Despite the use of a thorough screening technique, potentially relevant studies, including unpublished works or publications in languages other than English, may remain undiscovered. Thus, publication and linguistic bias cannot be eliminated. Moreover, the studies included in this review generally have smaller sample sizes, which may limit the generalizability of the findings. Additionally, there was an overrepresentation of male candidates in many studies, which limits our understanding of the efficacy of these supplements across both genders. A notable limitation is that the current review did not specifically evaluate whether the effects of herbal and amino acid supplements differ between individuals with ADHD alone and those with ADHD accompanied by psychiatric comorbidities. This limitation stems from the fact that the majority of the included studies failed to report or stratify results based on comorbid conditions (e.g., anxiety, depression, or oppositional defiant disorder), instead concentrating on ADHD symptomatology within broadly defined populations. For example, studies such as [17,19] demonstrated improvements in ADHD symptoms with *Bacopa monnieri*, yet did not analyze subgroups with comorbidities. Likewise, trials investigating L-theanine and caffeine did not consider potential interactions with comorbid psychiatric conditions. Future research should prioritize subgroup analyses to determine whether these supplements exert differential effects on individuals with ADHD alone compared to those with comorbidities, as this could inform more personalized treatment strategies. This gap highlights the necessity for more detailed reporting of comorbidities in clinical trials of ADHD interventions. Considering all these factors highlights the need for further research that better represents more diverse populations. Finally, a significant methodological challenge that may be encountered is the potential for publication overlap. This occurs when combining data from multiple reviews that include the same publications.

## 5. Conclusions

This systematic review evaluated the efficacy of five supplements that showed potential in the effective management of ADHD symptoms: L-theanine, caffeine, *Ginkgo biloba* L., and *Bacopa monnieri* either as standalone treatments or alongside stimulant medications. *Bacopa monnieri* has emerged as the most promising option for improving core ADHD symptoms such as inattention, hyperactivity, and impulsivity. L-theanine improved sleep efficiency but had no impact on ADHD symptoms. Caffeine was found to be ineffective in improving symptoms of ADHD, but it had a synergistic effect with L-theanine in enhancing sustained attention and cognitive performance. *Ginkgo biloba* L. moderately reduced inattention but was less effective than methylphenidate; however, it had fewer side effects, making it a possible alternative for those sensitive to stimulants. *Bacopa monnieri* consistently improved attention and cognitive function across multiple studies. While certain supplements, such as *Bacopa monnieri* may serve as promising complementary management options for ADHD under clinical guidance, their use as the sole treatment is not recommended. Further extensive studies are needed to evaluate the potential of these supplements as adjunctive therapies, their optimal dosages, and the long-term safety of their use.

## Figures and Tables

**Figure 1 nutrients-17-01482-f001:**
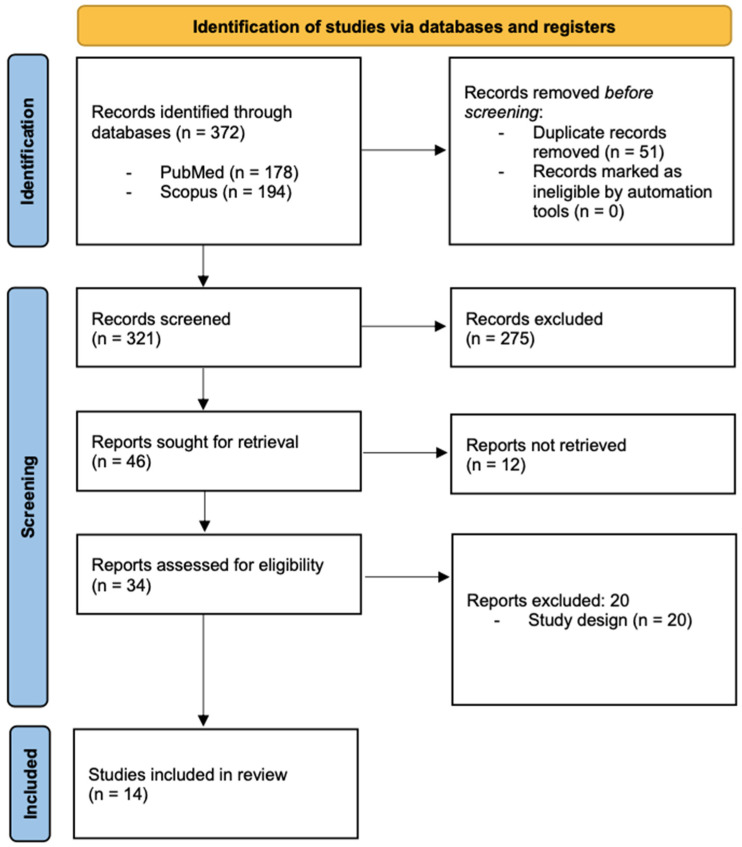
PRISMA flow diagram.

**Figure 3 nutrients-17-01482-f003:**
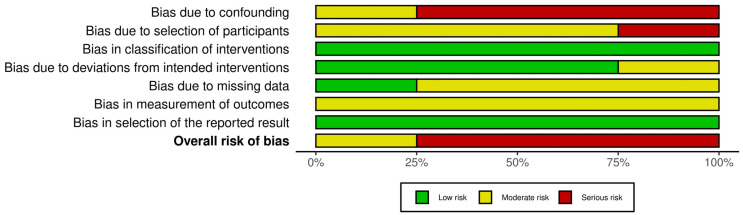
The results of the ROBINS-I bias assessment for cross-sectional and open-label studies [12,19,20,22] present risk-of-bias judgments across seven domains (color-coded as green [low], yellow [moderate], or red [serious risk]) and their overall classifications.

## Data Availability

The original contributions presented in the study are included in the article, further inquiries can be directed to the corresponding authors.

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
