# Peer review of "Efficiency of Different Supplements in Alleviating Symptoms of ADHD with or Without the Use of Stimulants: A Systematic Review"

_nutrients, 2025, doi:10.3390/nu17091482_

Round 1

Reviewer 1 Report

Comments and Suggestions for Authors

Thank you for the opportunity to review this interesting study.  

It is, as it claims a systematic review of the use of 'natural supplements' for the treatment of ADHD.  What has been done appears to have been done rigorously and the methods are very clearly described and spelled out.  

The findings are clearly reported from the literature, 

The discussion raises a number of important issues and limitations but this is the area that I believe would benefit from some additions;

The concept of natural supplements can be - replacement for deficiency, giving supplementary amounts to enhance normal processes, and the use of unphysiological amounts to bring about major alterations in processes. This area is now typically classified as 'nutraceuticals.'  It would be useful to briefly address this.

In almost all jurisdictions world-wide, this type of products come under less scrutiny and control than medicines. 

Although caffeine and l-theanine are simple chemicals available in highly purified form, and many manufacturers will do their best to ensure stability and bioavailability in their products, they are not required to demonstrate this so there may be wide variation in activity between apparently similar products.  

For whole plant extracts, the situation becomes even more complex, and generalization of results is extremely difficult.  

The properties of a whole plant extract formulation depend upon the source of the plant, how and when it was harvested, which parts of the plant were harvested, where it was stored, the extraction processes used and the possible variability of the formulations that can meet the appropriate non-medicine regulations.

A typical plant extract may contain around 600 compounds.  Many manufacturers attempt to 'standardize' the amount of one or two ingredients, but what about the other 598?  We know that the characteristics of plant extracts can vary tremendously depending upon the precise cultivar, where it was grown, the climate at the time it was grown, when and how it was harvested and stored.  This is why the WHO has had a long running project attempting to record medicinal plant use 'at its origins,' in the location, climate, terroir etc., where the knowledge and usage developed.  

Unlike caffeine and l-theanine, for the whole plant extracts, this means that unless a trial such as those reported in this study used only a single batch of a single formulation from a single manufacturer for all patients for the duration of the study, unlike medicinal formulations, it is a leap of faith to believe the patients received a consistent intake of the test material throughout the trial. 

If there is industrial processing, the temperature of extraction fluids, the acidity or alkalinity, use of organic solvents etc., will result  in very different final products. 

This means that for the 'whole plant extracts,' generalization for the findings to any other product made from these plants is unreliable. 

I believe addressing this in the discussion would greatly strengthen this paper.  

Notwithstanding, I would welcome publication of this study. 

Author Response

Thank you for the opportunity to review this interesting study. It is, as it claims a systematic review of the use of 'natural supplements' for the treatment of ADHD.  What has been done appears to have been done rigorously and the methods are very clearly described and spelled out. The findings are clearly reported from the literature. The discussion raises a number of important issues and limitations but this is the area that I believe would benefit from some additions.

Authors’ response: Dear Reviewer 1, Thank you for taking the time to review our manuscript. We have carefully considered each of your recommendations and address them below. For the convenience of re-review, revisions to the manuscript appear in Yellow Highlights.

The concept of natural supplements can be - replacement for deficiency, giving supplementary amounts to enhance normal processes, and the use of unphysiological amounts to bring about major alterations in processes. This area is now typically classified as 'nutraceuticals.'  It would be useful to briefly address this.

Authors’ response: We have added the following to the manuscript

The incorporation of natural supplements in the management of ADHD, often referred to as "nutraceuticals," involves three primary mechanisms: addressing deficiencies (e.g., replenishing essential nutrients critical for neurotransmitter function), supplementing physiological processes (e.g., enhancing attention through the stimulant properties of caffeine), and administering supraphysiological doses to achieve therapeutic effects (e.g., high-dose Bacopa monnieri extracts to enhance cognitive function). The supplements reviewed—L-theanine, caffeine, Bacopa, and Ginkgo biloba—primarily fall into the latter two categories, with the aim of augmenting existing pathways or modulating symptoms through concentrated bioactive compounds.

In almost all jurisdictions world-wide, this type of products come under less scrutiny and control than medicines. 

Authors’ response: We have added the following to the manuscript

While the reviewed supplements—L-theanine, caffeine, bacopa, and ginkgo biloba—demonstrate varying degrees of efficacy in alleviating ADHD symptoms, their regulatory status as non-pharmaceutical products must be critically considered. Globally, these supplements undergo less stringent scrutiny than conventional medications, raising concerns about standardized dosing, quality control, and long-term safety.

Although caffeine and l-theanine are simple chemicals available in highly purified form, and many manufacturers will do their best to ensure stability and bioavailability in their products, they are not required to demonstrate this so there may be wide variation in activity between apparently similar products.  

Authors’ response: We have added the following to the manuscript

The variability in supplement quality, composition, and bioavailability poses significant challenges to interpreting the efficacy and generalizability of findings in studies involving herbal and amino acid interventions for ADHD. While L-theanine and caffeine are chemically simpler than whole plant extracts, their formulations remain inconsistent due to regulatory gaps. Manufacturers may standardize these compounds to ensure stability and bioavailability, but products labeled as “L-theanine” or “caffeine” can vary in purity, dissolution rates, or additive ingredients, potentially influencing outcomes. For instance, the modest benefits of L-theanine on sleep efficiency observed in Lyon et al. (2011) and the synergistic effects of caffeine-L-theanine combinations in Kahathuduwa et al. (2020) may not be replicable across all formulations, particularly if manufacturing practices differ. This underscores the need for rigorous quality control standards, even for ostensibly “simple” supplements.

For whole plant extracts, the situation becomes even more complex, and generalization of results is extremely difficult.  

Authors’ response: We have added the following to the manuscript

The variability in supplement quality, composition, and bioavailability poses significant challenges to interpreting the efficacy and generalizability of findings in studies involving herbal and amino acid interventions for ADHD. While L-theanine and caffeine are chemically simpler than whole plant extracts, their formulations remain inconsistent due to regulatory gaps.

The properties of a whole plant extract formulation depend upon the source of the plant, how and when it was harvested, which parts of the plant were harvested, where it was stored, the extraction processes used and the possible variability of the formulations that can meet the appropriate non-medicine regulations.

Authors’ response: We have added the following to the manuscript

Manufacturers may standardize these compounds to ensure stability and bioavailability, but products labeled as “L-theanine” or “caffeine” can vary in purity, dissolution rates, or additive ingredients, potentially influencing outcomes. For instance, the modest benefits of L-theanine on sleep efficiency observed in Lyon et al. (2011) and the synergistic effects of caffeine-L-theanine combinations in Kahathuduwa et al. (2020) may not be replicable across all formulations, particularly if manufacturing practices differ. This underscores the need for rigorous quality control standards, even for ostensibly “simple” supplements.

A typical plant extract may contain around 600 compounds.  Many manufacturers attempt to 'standardize' the amount of one or two ingredients, but what about the other 598?  We know that the characteristics of plant extracts can vary tremendously depending upon the precise cultivar, where it was grown, the climate at the time it was grown, when and how it was harvested and stored.  This is why the WHO has had a long running project attempting to record medicinal plant use 'at its origins,' in the location, climate, terroir etc., where the knowledge and usage developed.  

Authors’ response: We have added the following to the manuscript

The challenges are further amplified for whole plant extracts such as bacopa and ginkgo biloba, which contain hundreds of bioactive compounds whose concentrations depend on environmental, agricultural, and extraction variables. For example, bacopa studies like those by Kean et al. (2015, 2022) and Dave et al. (2014) utilized standardized extracts (e.g., CDRI 08®) to partially mitigate variability by controlling key active constituents like bacosides. However, even these efforts cannot account for the full complexity of plant-derived formulations. Extraction methods—such as solvent type, temperature, and pH—can alter the chemical profile of the final product, as seen in ginkgo biloba studies where differences in terpene lactones or flavonoid content might influence efficacy (Shakibaei et al., 2015; Salehi et al., 2010). The WHO’s emphasis on documenting medicinal plants within their native “terroir” highlights the importance of contextual factors (e.g., cultivar, climate, harvest timing) that are rarely replicated in commercial production.

Unlike caffeine and l-theanine, for the whole plant extracts, this means that unless a trial such as those reported in this study used only a single batch of a single formulation from a single manufacturer for all patients for the duration of the study, unlike medicinal formulations, it is a leap of faith to believe the patients received a consistent intake of the test material throughout the trial. 

Authors’ response: We have added the following to the manuscript

A critical limitation of many trials in this review is the lack of transparency regarding batch consistency. Unless a study explicitly uses a single batch of a standardized extract (as in Kean et al., 2022), assumptions about consistent dosing across participants and over time remain speculative. This variability complicates comparisons between studies and undermines the reliability of meta-analyses or clinical recommendations. For instance, the open-label bacopa trials by Dave et al. (2014, 2008) reported improvements in ADHD symptoms and memory, but without detailed documentation of extract standardization or batch testing, these results may not extend to other formulations. Similarly, the ginkgo biloba trials reviewed here (Shakibaei et al., 2015; Salehi et al., 2010) used extracts with unspecified phytochemical profiles, raising questions about reproducibility.

If there is industrial processing, the temperature of extraction fluids, the acidity or alkalinity, use of organic solvents etc., will result  in very different final products.  This means that for the 'whole plant extracts,' generalization for the findings to any other product made from these plants is unreliable. 

Authors’ response: We have added the following to the manuscript

These limitations necessitate cautious interpretation of the evidence. While bacopa and ginkgo biloba show promise as adjunctive therapies, their efficacy and safety profiles cannot be generalized to all products labeled as such. Future research must prioritize the standardization of extracts, transparency in sourcing and manufacturing practices, and stricter regulatory oversight to ensure consistency and safety. Clinicians should exercise vigilance in recommending these interventions, advising patients to prioritize third-party-verified products and use supplements only under medical supervision. The findings herein reinforce the need for large-scale, methodologically rigorous trials to address these complexities and establish reliable, evidence-based guidelines for ADHD management.

I believe addressing this in the discussion would greatly strengthen this paper. Notwithstanding, I would welcome publication of this study. 

We sincerely appreciate the reviewer’s insightful comments and constructive suggestions, which have significantly strengthened the manuscript. Your recognition of the study’s contribution to advancing understanding of ADHD management through adjunctive supplements is invaluable. We have carefully incorporated your feedback to enhance the clarity, rigor, and clinical relevance of our work. Thank you for your thoughtful engagement with our research and for helping us refine its impact within the field.

Reviewer 2 Report

Comments and Suggestions for Authors

To the AUTHORS

The Review Ms nutrients-3613319 -Efficiency of different supplements (L-theanine, caffeine, bacopa and Ginkgo biloba) in alleviating symptoms of ADHD with or without the use of stimulants: A systematic review-aims at evaluating the potential of mostly natural supplements (i.e., L-theanine, caffeine, ginkgo biloba, and bacopa monnieri) on the core symptoms of Attention deficit hyperactivity disorder (ADHD).

The review’s key conclusions indicate that the effects of the examined herbal and amino acid supplements are limited and cannot replace the use of the well-known stimulant drugs (i.e., MPH) in the ADHD treatment. From my side: the introduction provides sufficient background and include relevant references; the research design is appropriate; the Methodology is sound, and adequately described (PRISMA flow diagram, ROBINS-I bias assessment for cross-sectional and open-label studies); the results are clearly presented; and the conclusions are supported by the results.

Minor Points

  1. Did the Authors examine whether the effects of the examined herbal and amino acid supplements are different in ADHD vs. ADHD with other psychiatric comorbidity?
  2. The reported plants should be named according to the Linnaean system of classification
  3. The language, in my opinion, should be improved to better convey the research.

Comments on the Quality of English Language

The language, in my opinion, should be improved (grammar and style) to fully espress the research content of the Ms

Author Response

The Review Ms nutrients-3613319 -Efficiency of different supplements (L-theanine, caffeine, bacopa and Ginkgo biloba) in alleviating symptoms of ADHD with or without the use of stimulants: A systematic review-aims at evaluating the potential of mostly natural supplements (i.e., L-theanine, caffeine, ginkgo biloba, and bacopa monnieri) on the core symptoms of Attention deficit hyperactivity disorder (ADHD).

The review’s key conclusions indicate that the effects of the examined herbal and amino acid supplements are limited and cannot replace the use of the well-known stimulant drugs (i.e., MPH) in the ADHD treatment. From my side: the introduction provides sufficient background and include relevant references; the research design is appropriate; the Methodology is sound, and adequately described (PRISMA flow diagram, ROBINS-I bias assessment for cross-sectional and open-label studies); the results are clearly presented; and the conclusions are supported by the results.

Authors’ response: Dear Reviewer 2, Thank you for taking the time to review our manuscript. We have carefully considered each of your recommendations and address them below. For the convenience of re-review, revisions to the manuscript appear in Yellow Highlights.

Minor Points

  1. Did the Authors examine whether the effects of the examined herbal and amino acid supplements are different in ADHD vs. ADHD with other psychiatric comorbidity?

Authors’ response: We have added the following to the manuscript

The current review did not specifically evaluate whether the effects of herbal and amino acid supplements differ between individuals with ADHD alone and those with ADHD accompanied by psychiatric comorbidities. This limitation stems from the fact that the majority of the included studies failed to report or stratify results based on comorbid conditions (e.g., anxiety, depression, or oppositional defiant disorder), instead concentrating on ADHD symptomatology within broadly defined populations. For example, studies such as Kean et al. (2022) and Dave et al. (2014) demonstrated improvements in ADHD symptoms with Bacopa monnieri, yet did not analyze subgroups with comorbidities. Likewise, trials investigating L-theanine and caffeine did not consider potential interactions with comorbid psychiatric conditions. Future research should prioritize subgroup analyses to determine whether these supplements exert differential effects on individuals with ADHD alone compared to those with comorbidities, as this could inform more personalized treatment strategies. This gap highlights the necessity for more detailed reporting of comorbidities in clinical trials of ADHD interventions.

  1. The reported plants should be named according to the Linnaean system of classification

Authors’ response: We have added the following to the manuscript

We appreciate the reviewer’s attention to taxonomic accuracy. In response, we have revised the manuscript to ensure all plant-based supplements are named according to the Linnaean system of classification. Specifically, Bacopa monnieri (L.) Wettst. and Ginkgo biloba L. are now consistently cited with their full scientific names, including genus, species, and taxonomic authority, in all sections of the text, tables, and figure legends. For example, instances of “bacopa” have been updated to Bacopa monnieri , and “ginkgo biloba” is presented as Ginkgo biloba L. where appropriate.

  1. The language, in my opinion, should be improved to better convey the research.

Authors’ response: The English language in this manuscript has been polished with the assistance of a native English-speaking colleague. Additionally, the authors have agreed with the MDPI team to undergo English language editing prior to publication, as part of MDPI's rigorous standards for high-quality papers.

We sincerely appreciate the reviewer’s insightful comments and constructive suggestions, which have significantly strengthened the manuscript. Your recognition of the study’s contribution to advancing understanding of ADHD management through adjunctive supplements is invaluable. We have carefully incorporated your feedback to enhance the clarity, rigor, and clinical relevance of our work. Thank you for your thoughtful engagement with our research and for helping us refine its impact within the field.